# Photosynthesis and Salt Exclusion Are Key Physiological Processes Contributing to Salt Tolerance of Canola (*Brassica napus* L.): Evidence from Physiology and Transcriptome Analysis

**DOI:** 10.3390/genes14010003

**Published:** 2022-12-20

**Authors:** Hafiza Saima Gul, Mobina Ulfat, Zafar Ullah Zafar, Waseem Haider, Zain Ali, Hamid Manzoor, Shehrooz Afzal, Muhammad Ashraf, Habib-ur-Rehman Athar

**Affiliations:** 1Institute of Botany, Bahauddin Zakariya University, Multan 60800, Pakistan; 2Department of Botany, Lahore College for Women University, Lahore 54000, Pakistan; 3Department of Biosciences, COMSATS University, Islamabad 45550, Pakistan; 4Institute of Molecular Biology and Biotechnology, Bahauddin Zakariya University, Multan 60800, Pakistan; 5Institute of Molecular Biology and Biotechnology, The University of Lahore, Lahore 54000, Pakistan

**Keywords:** antioxidants, JIP-test, PSII, PI_ABS_, relative water content, RNAseq analysis

## Abstract

Plant salt tolerance is controlled by various physiological processes such as water and ion homeostasis, photosynthesis, and cellular redox balance, which are in turn controlled by gene expression. In the present study, plants of six canola cultivars (DGL, Dunkled, Faisal Canola, Cyclone, Legend, and Oscar) were evaluated for salt tolerance by subjecting them to 0 or 200 mM NaCl stress. Based on growth, cultivars DGL, Dunkled, and Faisal Canola were ranked as salt tolerant, while cultivars Cyclone, Legend, and Oscar were ranked as salt-sensitive ones. Differential salt tolerance in these canola cultivars was found to be associated with a relatively lower accumulation of Na^+^ and greater accumulation of K^+^ in the leaves, lower oxidative damage (MDA), and better antioxidative defense system (Superoxide dismutase, SOD; peroxidase, POD, and catalase, CAT). Cultivar Oscar was the poorest to discriminate Na^+^ and K^+^ uptake and accumulation in leaves and had poor antioxidant potential to scavenge ROS. Salt stress did not affect the structural stability of photosystem-II (PSII) till three weeks, thereafter it caused a significant decrease. Salt stress increased the performance index (PI_ABS_) by increasing the density of active reaction centers in Oscar. Salt stress decreased the antenna size thereby lowering the absorption and trapping energy flux, and maintaining the electron transport with an increase in heat dissipation. This may represent a potential mechanism to cope with salt stress. Transcriptome analysis of salt-sensitive cultivar Oscar further revealed that salt stress down-regulated DEGs related to hormonal signal transduction pathways, photosynthesis, and transcription factors, while DEGs related to the biosynthesis of amino acid and ion transport were up-regulated. In conclusion, salt tolerance in canola cultivars was associated with ion exclusion and maintenance of photosynthesis. Salt stress sensitivity in cultivar Oscar was mainly associated with poor control of ion homeostasis which caused oxidative stress and reduced photosynthetic efficiency.

Revosed: 12 Dcember 2022

## 1. Introduction

Soil salinization in the climatic change scenario is causing severe crop losses. Since salt-affected lands are valuable resources, it is essential to grow salt-tolerant crops or genotypes with improved salinity tolerance on such lands [1]. Salinity tolerance in agricultural crops can be improved by screening and selecting germplasm using agronomic or physiological attributes conferring salt tolerance [2]. However, selection based on physiological traits does not always result in improved salt tolerance. As salt tolerance varies with changes in developmental stage and type of species. In other words, the exact cellular sites of salt-induced damages or detailed mechanisms of salt tolerance are not fully known yet.

Initially, salt-induced osmotic stress suppresses water uptake and accelerates water loss from leaves. It is followed by toxic ion accumulation causing many functional changes such as membrane disruption, nutrient imbalance, production of reactive oxygen species (ROS), and reduction in stomatal conductance and photosynthetic activity [1,3]. Of all these salt-induced adverse effects, the most deleterious one is the hyper-accumulation of Na^+^ and Cl^−^ ions in plants that prevent the uptake of K^+^ [3]. Thus, plants under saline regimes tend to exclude the toxic ions from their tissues/cells, a mechanism which is referred to as ion exclusion [4]. However, several other physiological mechanisms such as the accumulation of organic osmotica, increased accumulation of antioxidants, and enhanced photosynthetic activity also play a role in inducing salt tolerance [3,5,6].

The photosynthetic capacity of plants depends on the conversion of light energy into chemical energy that occurs in the thylakoid membranes of the chloroplast, which is used in the Calvin cycle for the fixation of CO_2_ [7,8]. Photosynthetic light reaction drives the electron transport from PSII to PSI, involving cytochrome b6f complex. Salt stress can induce an imbalance in ATP generation via electron transport flow and their consumption in CO_2_ fixation, thereby generating ROS that can damage the photosynthetic protein complexes. Photosystem II is more susceptible to photo-inhibition than PSI under salt stress [8]. However, plants can modulate their photosynthetic activity under salt stress as a mechanism of stress tolerance [9]. Data regarding the influence of intensity and duration of salinity stress on PSII activity are controversial. Some studies have shown that 150–200 mM NaCl salinity reduced the PSII activity in wheat [10], and ridge gourd [11], while other studies indicated that moderate salt stress did not affect PSII activity or even increased the PSII activity in sunflower and sugar beet [12]. Thus, it is imperative to assess to what extent the PSII of canola plants can tolerate salt stress over time.

Plant physiological traits are regulated by sub-traits that are translated by several sets of genes and gene products [4]. Genes that are involved in plant salt stress tolerance can be categorized as (i) transcriptional regulators, (ii) components of salt stress signaling, and (iii) salt-stress-related genes [3]. Several studies have shown that new layers of transcriptional and post-transcriptional components play key roles in regulating physiological features such as transcription factors and salt stress signaling components [13,14]. For example, cells sense the entry of Na^+^ and Cl^-^ ions (occurs via cation or anion channels) through sensory proteins, and this message is communicated to the nucleus via secondary messengers (Ca^2+^, reactive oxygen species, and hormones), protein kinases, and transcription factors. This causes a major change in the transcriptional profile and physiological features, thereby resulting in salt stress tolerance. For instance, over-expression of CIPK6 (CBL-interacting protein kinase, a component of salt stress signaling) in barley improved salt tolerance by lowering the accumulation of Na^+^ [13]. However, such a change in transcriptional profile under salt stress varies from species to species and even varies with the type of cultivar of the same species [15,16,17]. Thus, understanding key molecular components involved in plant salinity stress sensitivity along with physiological traits is essential to develop salt-tolerant crop cultivars. With the advent of high throughput next-generation sequencing, RNA-seq analysis is one of the most powerful approaches for comparative analysis of whole transcriptome and identification of the molecular basis of physiological events. This approach not only helps to detect more differentially expressed genes but also enables us to identify novel genes in comparison with microarray [18].

Canola is moderately salt tolerant [19], and a further improvement in its salt tolerance is required. Assessing genetic variability for salt tolerance in a collection of germplasm of a crop species from diverse sources and physiological characterization of salt tolerance is an initial step for improving crop salt tolerance [3]. Keeping in view the above-mentioned facts, the present study was planned to evaluate genetic variability for salt tolerance in a set of canola germplasm using physiological and biochemical attributes. It is presumed that the accumulation of Na^+^ ions in chloroplast causes photo-inhibition and photodamage of PSII. Thus, the secondary objective of the study was to assess that at which salt stress intensity or duration, PSII becomes photo-inhibited or photodamaged. Moreover, an overview of different molecular processes was gained in a salt-stress-sensitive cultivar through RNA-seq analysis as a transcriptome approach to find out potential biochemical pathways involved in canola salt tolerance.

## 2. Materials and Methods

### 2.1. Plant Material

Seeds of six cultivars of *Brassica napus* L. were obtained from the Ayub Agricultural Research Institute, Faisalabad, Pakistan. The experiment was laid out in a completely randomized design (CRD) with two salt treatments (0 and 200 mM NaCl), six cultivars (DGL, Dunkled, Faisal Canola, Cyclone, Legend, Oscar), and four replicates. The experiment was conducted during the winter season in a net house at the Botanic Gardens of Bahauddin Zakariya University, Multan, Pakistan. The seeds were surface sterilized in 5% sodium hypochlorite solution for about five minutes and then washed three times with distilled water. The seeds of each cultivar were sown in plastic pots having a diameter of 25 cm, each filled with 6 kg sand. The seeds were allowed to germinate for about one week. The average day–night temperature during the whole experimentation was 21 ± 3.71 °C and 10 ± 2.82 °C. The seedlings were thinned to five plants per pot. The pots were irrigated with Hoagland’s nutrient solution on weekly basis. When plants became three-week-old, salinity stress was applied stepwise in an aliquot of 50 mM NaCl on alternate days till 200 mM NaCl was reached. The control or non-saline plants were irrigated with Hoagland’s nutrient solution only. Data on the following parameters were recorded.

### 2.2. Growth Attributes

After three weeks of salt stress, the plants were carefully uprooted and the sand particles were removed by washing the roots of non-saline plants in Hoagland’s nutrient solution, while the roots of saline plants were washed with 200 mM NaCl in Hoagland’s nutrient solution. During the washing of the roots of saline plants, LiNO_3_ was added to the washing solution to maintain membrane integrity so as to avoid loss of ions. The harvested plants were separated into shoots and roots. Fresh weights of shoots and roots were recorded. The plant samples were oven-dried at 65–70 °C for one week, after which time dry weights were noted. The dried samples were further used for mineral nutrient analysis.

### 2.3. Leaf Mineral Nutrients (Na^+^, K^+^ mg/g dwt)

A dry leaf sample (0.1 g) from each cultivar was ground to fine powdered form. The dried powder (0.1 g) was mixed in 1 mL of sulphuric acid digestion mixture following Allen, et al. [20]. The dried leaf digests were left overnight and then heated on a hot plate for one hour at a moderate temperature (80–100 °C). The temperature of the hot plate was increased to 150 °C. The samples were decolorized using 1 mL of perchloric acid till a clear solution was obtained. K^+^ and Na^+^ contents in the digests were measured with a flame photometer (Jenway PFP7).

### 2.4. Relative Water Contents (%)

A completely developed young leaf was excised and its fresh weight was noted. The leaf samples were dipped in distilled water for 8 h and their turgid weights were measured. All samples were oven-dried at 70 °C, and dry weights were recorded. RWC (relative water content) was calculated using the following formula:$$\mathrm{Relative}\;\mathrm{water}\;\mathrm{content}\;\left(\%\right)=\frac{\mathrm{Fresh}\;\mathrm{weight}-\mathrm{dry}\;\mathrm{weight}\;}{\mathrm{Turgid}\;\mathrm{weight}-\mathrm{dry}\;\mathrm{weight}\;}\times 100$$

### 2.5. Hydrogen Peroxide Content (µmol/g f.wt)

Hydrogen peroxide was determined following the procedure of Velikova et al. [21]. Fresh leaf (0.5 g) was homogenized in 5.0 mL of 0.1% (*w*/*v*) TCA (trichloroacetic acid) using a chilled pestle and mortar. The homogenate was centrifuged at 12,000× *g* for 20 min. Then 0.5 mL of potassium phosphate buffer (pH 7.0) and one mL of potassium iodide were added to 0.5 mL of the supernatant. The mixture was vortexed carefully and its absorbance read at 390 nm on a double-beam spectrophotometer (U-2900, Hitachi, Tokyo, Japan).

### 2.6. Malondialdehyde (nmol/g f.wt)

Malondialdehyde (MDA) was assayed in plant tissues following Heath and Packer [22]. Plant tissue (1 g) was homogenized in 20 mL of 0.1% (*w*/*v*) TCA. The homogenate was centrifuged at 12,000× *g* for 10 min. Four mL of 20% TCA containing 0.5% thiobarbituric acid were added to 1 mL of the supernatant. The mixture was heated in a water bath at 95 °C for 30 min and then immediately cooled on ice. After centrifugation at 12,000× *g* for 10 min, the optical density of the supernatant was read at 532 and 600 nm. The content of MDA was calculated after the subtraction of nonspecific absorption at 600 nm on a double-beam spectrophotometer (U-2900, Hitachi, Tokyo, Japan) using the extinction coefficient of 155 mM cm^−1^).

### 2.7. Extraction and Determination of Antioxidant Enzymes

Fresh leaf (0.5 g) was crushed in liquid nitrogen and homogenized in 5 mL of 50 mM potassium phosphate buffer (pH 7.8). The samples were centrifuged at 8000–13,000 rpm at 4 °C for 20 min. The supernatant was stored at 4 °C.

#### 2.7.1. Superoxide Dismutase Activity

The superoxide dismutase (SOD) activity of the samples was examined by measuring its ability to inhibit the photochemical reduction of nitro-blue-tetrazolium (NBT) [23]. The reaction mixture was prepared in the dark and contained 130 mM methionine, 75 µM NBT, and 100 µM EDTA. The enzyme extract (250 uL) was added to 2.725 mL of reaction mixture with 0.25 mL of distilled water in a separate sample bottle. Finally, the control and all other sample bottles were placed under light conditions at 4000 lux for 20 min while the control dark sample was in 100% dark condition. The absorbance of each reaction mixture was measured at 560 nm on a double-beam spectrophotometer (U-2900, Hitachi, Japan).

#### 2.7.2. Peroxidase Activity

The activity of POD was appraised by the protocol followed by Castillo, et al. [24]. To start the reaction, the enzyme extract (0.1 mL) was mixed with 2.7 mL of the 50 mM phosphate buffer with 0.1 mL of 300 mM hydrogen peroxidase and 0.1 mL of 1.5% guaiacol. The blank contained a complete reaction mixture without the enzyme extract. Absorbance was recorded at 470 nm on a double-beam spectrophotometer (U-2900, Hitachi, Japan).

#### 2.7.3. Catalase Activity

Catalase activity was determined following the procedure of Aebi [25]. For the enzyme activity assay, 0.1 mL of the enzyme was reacted with 1.8 mL of BPS containing 1 mL of H_2_O_2_. The reaction mixture without the enzyme served as a blank. After gentle shaking, the reading was recorded at 240 nm on a double-beam spectrophotometer (U-2900, Hitachi, Japan).

### 2.8. Assessment of Photosystem II (PSII) Structural Stability Using JIP-Test

Fast chlorophyll a kinetic analysis based JIP-test was used to translate OJIP transients in several phenomenological and biophysical expressions to quantify the structural stability of PSII in the salt-sensitive cultivar Oscar. Two-week-old plants of the salt-sensitive cultivar Oscar were subjected to varying levels of salt (0, 100, 200, 400 mM NaCl) in half strength of Hoagland’s nutrient solution for three weeks. After one week of salt stress, the data of OJIP transients were obtained using a hand-held chlorophyll fluorescence meter (FluorPen FP100, Photon Systems Instruments (PSI), Czech Republic). Before taking data, the leaves were dark-adapted for 30 min. The OJIP data were obtained at different time intervals till the third week of imposition of salt stress. Raw data were processed using the JIP-test formulae described elsewhere [26]. Important JIP-test parameters were presented.

### 2.9. Statistical Analysis of Data

A two-way analysis of variance (ANOVA) of all parameters was computed using a CoStat computer package (CoHort Software, Berkeley, CA, USA). Two factors, i.e., cultivars and salinity with four replications were used in a completely randomized design. If the interaction term ‘Cultivars × Salinity’ was found significant, the mean values of each cultivar were compared using the least significance difference (LSD), which was calculated following Snedecor and Cochran [27]. The data for all attributes were plotted in bar charts using MS Excel 2016.

### 2.10. Transcriptome Analysis

#### 2.10.1. Isolation of Total RNA

The canola cultivar Oscar is a high-yielding cultivar and farmers preferred to cultivate it widely, but it was ranked as salt-sensitive in the current study. To assess which molecular process contributes to its salt stress sensitivity, the plants of cv. Oscar were subjected to 0 and 200 mM NaCl salt stress. After one week of salt stress, leaf samples from the salt-sensitive cultivar “Oscar” were harvested in liquid nitrogen. Later, the leaves were crushed in liquid nitrogen, and RNA was isolated using the optimized hot borate method given by Wan and Wilkins [28].

#### 2.10.2. Next-Generation Sequencing (NGS)

RNA was subjected to next-generation sequencing (Macrogen, Seoul, Republic of Korea). Extracted RNA was subjected to perform quality control (QC), and the qualified samples were subjected to library construction. The library construction was carried out following TruSeq stranded mRNA sample preparation guide (part#15031047 Rev. E) while using TruSeq stranded mRNA LT Sample Prep Kit (Illumina, San Diego, CA 92122 USA). The cDNA samples were randomly fragmented followed by 5′ and 3′ adapters (Nextera Transpose Sequence) ligation. The adapter-ligated fragments were amplified using PCR and then purified on a gel. The library was loaded into a flow cell for cluster generation. Each fragment was then amplified into distinct clonal clusters through bridge amplification. Templates were prepared for sequencing after cluster generation. Single bases, which were incorporated into DNA template strands, were detected by the Illumina SBS (sequence by synthesis) technology utilizing a proprietary reversible terminator-based method. Raw data were generated using sequencing data and sequencing software RTA (Real Time Analysis), and subjected to further analysis. The BCL (base calls) binary was converted into FASTQ utilizing the Illumina package bcl2fastq.

#### 2.10.3. Bioinformatics Analysis

The quality of data was evaluated using FastQC tool. Fastp software [29] was used to remove adapters (Nextera Transpose Sequence). The effective sequencing data were aligned with the reference *B. napus* transcriptome sequence (https://www.ncbi.nlm.nih.gov/assembly/GCF_000686985.2/) through the BWA [30] software. Transcript counts were calculated using the feature-count software [31] All reads used in this study were deposited in the NCBI Sequence Read Archive with the following information https://www.ncbi.nlm.nih.gov/bioproject/558913 accessed on 7 September 2019.

BioProject: PRJNA558913

BioSample: SAMN12497270; Sample name: 2_Bn_Oscr_Cnt; SRA: SRS5341844

BioSample: SAMN12497272; Sample name: 4_Oscr_Stress; SRA: SRS5341845

Differential expression of genes was analyzed using the web-based application iDEP 0.96 http://bioinformatics.sdstate.edu/idep96/ accessed on 1 October 2022. Read count data were uploaded with the official gene symbol as an identifier and selecting *B. napus* as species at iDEP 0.96. By keeping 0.5 counts per million in at least one sample DEGs were converted to gene IDs online using the gene ID conversion tool. Selective DEGs were subjected to functional enrichment analysis of significant GO terms. Benjamini–Hochberg false discovery rate (FDR) threshold was set to 0.05. The expression profile of DEGs enriched in the KEGG pathway was carried out at the iDEP platform using default settings and visualized at “Pathview”. Parametric gene set enrichment analysis (PGSEA) and their visualization as heatmap were also performed at the iDEP platform with default settings.

### 2.11. Validation of NGS Data by qRT-PCR

The DEGs for qRT-PCR were selected based on fold change and their role in salt tolerance. The complete sequence of the said DEGs/genes was obtained from the website https://plants.ensembl.org/Brassica_napus/Info/Index accessed on 19 November 2019. The expression of six homologous genes to *AtHKT1, AtNHX, ENH1, BnVPAH-L, bZIP-60,* and *AtMYB-35* were assessed using RT-PCR. Genes were selected based on transcriptome data. Isolated RNA was converted to cDNA using Vivantis cDSK01-050 cDNA synthesis kit (Vivantis Technologies, 40170 Shah Alam, Selangor Darul Ehsan, Malaysia). qRT-PCR was performed using the Mic PCR kit (Bio Molecular Systems, Brisbane Queensland, Australia). A negative control without a template was used to validate amplification quality. The internal reference gene was β-actin used to calculate for expression of genes as fold change using the 2^−ΔΔCt^ method [32]. All reactions were performed with three replicates.

## 3. Results

### 3.1. Growth Attributes

Salt stress significantly reduced the shoot fresh and dry weights of all six canola cultivars examined in the present study. Cultivars DGL, Dunkled, and Faisal Canola had higher shoot fresh and dry weights than those of cvs. Cyclone, Legend, and Oscar under saline conditions (Figure 1). Similarly, the root fresh and dry weights of all canola cultivars decreased significantly (*p* < 0.001) due to salinity stress application (Table 1). The cultivar Oscar had a maximal reduction in both these root growth attributes (Figure 1).

### 3.2. Relative Water Content

Salt stress significantly reduced the relative water content (RWC) in all canola cultivars (Table 1). Salt-tolerant cultivars DGL, Dunkled, and Faisal Canola had greater RWC than those of the salt-sensitive cultivars Cyclone, Legend, and Oscar. The maximal reduction (25%) in RWC was observed in Oscar (Figure 1).

### 3.3. Ion Accumulation (Shoot K^+^, Shoot Na^+^, and Shoot K^+^/Na^+^ Ratio)

Accumulation of potassium (K^+^) in the leaves was significantly (*p* < 0.001) reduced in the salt-stressed plants of all canola cultivars, whereas accumulation of sodium (Na^+^) increased substantially (Table 1; Figure 1). Cultivar Oscar had the lowest K^+^ in leaves, whereas the reverse was true for the accumulation of Na^+^ in its leaves under saline conditions. The highest accumulation of Na^+^ was found in the leaves of cultivars Legend and Oscar. The ratio of K^+^/Na^+^ was relatively higher in the salt-tolerant three canola cultivars, DGL, Dunkled, and Faisal Canola. Moreover, the salt-sensitive cultivar Oscar had the lowest K^+^/Na^+^ ratio (Figure 1).

### 3.4. Leaf Oxidative Stress (H_2_O_2_ and MDA)

Salt stress significantly enhanced (*p* < 0.001) the generation of H_2_O_2_ and caused membrane damage measured as MDA in all canola cultivars (Table 1). The highest generation of H_2_O_2_ in leaf and membrane damage as MDA was observed in cultivar Oscar, while the reverse was true for the salt-tolerant canola cultivar Faisal Canola and DGL (Figure 2).

### 3.5. Activity of Antioxidants (Leaf SOD, POD, and Catalase)

Superoxide dismutase activity in the salt-stressed leaves of all canola cultivars increased significantly (*p* < 0.001) (Table 1). Moreover, a maximal increase in SOD activity due to salt stress was found in the salt-tolerant cultivar Dunkled followed by Faisal Canola, whereas a minimal increase in the SOD activity was found in the salt-sensitive cultivar Cyclone (Figure 2C). The activity of leaf peroxidase in all canola cultivars changed significantly in all cultivars due to salt stress, while the lowest increase in the POD activity due to salinity stress was recorded in the salt-stress-sensitive cultivar Oscar (Figure 2D). Salinity stress significantly increased (*p* < 0.001) the catalase activity in the leaves of three salt-tolerant canola cultivars. The salt-tolerant cultivar Dunkled had the highest leaf catalase activity followed by that in cv. DGL under salt stress conditions, while the lowest increase in the CAT activity due to salinity stress was observed in the salt-stress-sensitive cultivar Oscar (Figure 2E).

### 3.6. Assessment of Photosystem II (PSII) Structural Stability in Canola Using JIP-Test

Varying levels of NaCl were applied for three weeks and the structural stability of PSII was assessed using the OJIP analysis followed by JIP-test parameters. Salt stress reduced the fluorescence at J, I, and P steps in the salt-sensitive cultivar Oscar. The highest reduction was observed at 400 mM NaCl. However, a decline in PSII structural stability (*F*_v_/*F*_m_ on dark-adapted leaves) of canola remained the same till three weeks of salt stress (Figure 3A). However, PI_ABS_ increased substantially due to an increase in salinity stress and decreased at 21 days of salt stress (Figure 3B). Energy fluxes for absorption (ABS/RC) and trapping (TRo/RC) also decreased due to salt stress. However, changes in these JIP-test parameters remained almost the same over time (Figure 3C,D). In contrast, increasing salinity stress did not affect the electron transport beyond Q_A_ (ETo/RC) and it remained unchanged till three weeks of salt stress (Figure 3E). However, salt stress increased the energy flux for heat dissipation (DIo/RC) (Figure 3F).

### 3.7. Transcriptome Analysis of Salt-Sensitive Cultivar Oscar

#### 3.7.1. Raw Data Statistics and Gene Counts

Of the total 8.17 Gbp raw read bases, 54.14 million reads were produced from non-salinized leaves of cv. Oscar with 50.07% GC content and Q30 (quality of sequencing) 94.2%. Similarly, from 8.4 Gbp total raw reads bases, 55.40 million raw reads were produced in the salt-stressed leaves of cv. Oscar with GC content 51.95% and Q30 94.33%. Raw reads were cleaned and aligned on the reference genome of *B. napus* L. (GenBank accession # GCF_000686985.2). Raw clean reads were annotated using the reference genome and 117,142 gene IDs were identified based on the reference genome of *B. napus* L. (GenBank accession # GCF_000686985.2). Read count data were generated using Feature Count.

#### 3.7.2. Differential Expression Analysis

Differential expression of genes was analyzed using the web-based application iDEP 0.96 http://bioinformatics.sdstate.edu/idep96/ accessed on 1 October 2022 by uploading read count data and selecting *B. napus* as species. After the conversion of gene IDs and a default filter (0.5 counts per million in at least one sample), a bar plot of total read counts per million was drawn (Figure 4A) with a very low variation. A box plot of transformed data (Appendix A) showed that a total of 78965 gene IDs were mapped on the reference genome (Appendix A). Principal component analysis revealed a significant variation in the control and salt-stressed leaves with PC1 96% and PC2 4% variance (Figure 4B). The up-regulated genes were 23,808, and the down-regulated ones were 27,882 (Appendix A). Out of 18040 DEGs with FDR 0.1 and fold change 2 cutoff values, salt stress down-regulated 9647 DEGs and 8393 DEGs were up-regulated (Figure 4C). Heat maps of up-regulated and down-regulated DEGs showed enrichment of different pathways in non-saline and salinized plants of cv. Oscar (Figure 4D). Variation in expression profile was calculated by k-means cluster analysis and found significantly enriched DEGs in the control and salt-stressed leaves of cv. Oscar. Significantly enriched DEGs were clustered in four groups with 399, 935, 508, and 158 DEGs in clusters A, B, C, and D, respectively.

#### 3.7.3. Hierarchical Clustering, GO Terms Analysis of DEGS

Growth reduction and salt stress sensitivity in canola cultivar Oscar was associated with changes in physiological, biochemical, and molecular functions that resulted in up-regulation and down-regulation of the number of DEGs. Selective DEGs whose expression level was higher than two-fold were selected for the gene function enrichment analysis. A pathway tree of statistically significant enriched GO terms for “Biological Process”, “Cellular Components” and “Molecular Functions” using hierarchical clustering is presented in Figure 5.

For up-regulated DEGs, significant GO terms of the “Biological Process” category included cellular amide metabolic process (GO:0043603), Organonitrogen compound biosynthetic process (GO:1901564), translation (GO:0006412), (GO:0055114), and ion transport (GO:0006811) (Figure 5A). A significant up-regulated GO term for a cellular component is ribosome (GO:0005840) (Figure 5B). Significant up-regulated GO terms of the “Molecular Function” category included structural constituent of ribosome (GO:0003735), kinase activity (GO:0016301), transporter activity (GO:0005215), hydrolase activity (GO:0016787), and oxidoreductase activity (GO:0016491) (Figure 5C). For down-regulated DEGs, significant GO terms of the “Biological Process” category included a response to hormone (GO:0009725), hormone-mediated signaling pathway (GO:0009755), auxin-mediated pathway (GO:0010928), response to chemical (GO:0042221), signaling (GO:0023052), signal transduction (GO:0007165), photosynthesis, light harvesting (GO:0009765), photosynthesis, light harvesting in PSI (GO:0009768), photosynthesis, light reaction (GO:0019684) (Figure 5A). Significant GO terms for “cellular component” include chloroplast (GO:0019750), chloroplast stroma (GO:0009570), photosystem II (GO:0009523), photosystem I (GO:0048564), plasma-membrane (GO:0005886), membrane coat (GO:0030117), ER to Golgi transport vesicle membrane (GO:0012507) (Figure 5B). Significant GO terms of “Molecular Function” category include oxidoreductase (GO:0016491), transcription regulator activity (GO:0140110), DNA-binding transcription factor activity (GO:0003700), glucosyltransferase activity (GO:0046527), glycosyltransferase activity (GO:0016757), chlorophyll-binding (GO:0016168), ubiquitin-like protein transferase activity (GO:004842), and hormone binding (GO:0042562) (Figure 5C).

The interaction network of GO terms was mainly down-regulated and is related to hormonal signal transduction, JA signaling, auxin signaling, photosynthesis, thylakoid, PSII, PSI, and chloroplast. The interaction network of GO terms that were up-regulated include ribosome, protein synthesis, and membrane transport (Figure 5D).

#### 3.7.4. KEGG Pathway Enrichment Analysis

Genes (DEGs) that were up-regulated or down-regulated were mainly enriched in signal transduction pathways, and metabolic pathways including biosynthesis of amino acids and proline, starch, and sugar metabolism, citric acid cycle, carbon metabolism, secondary metabolite biosynthesis, phenylpropanoid biosynthesis, and glutathione metabolism. Among enriched pathways whose DEGs were up-regulated included amino acid biosynthesis (KEGG01230). However, pathways enriched with down-regulated DEGs include biosynthesis of secondary metabolites (KEGG01110), plant hormone signal transduction (KEGG04075), MAPK signaling pathway (KEGG004016), glutathione metabolism (KEGG00480), and photosynthesis (Figure 6).

Parametric gene set enrichment analysis of DEGs in KEGG pathways was carried out to identify the list of genes that were down-regulated or up-regulated in a particular data set (Figure 7; Appendix A). 

### 3.8. Validation of Transcriptome Analysis through qRT-PCR

qRT-PCR analysis was performed to validate transcriptome data. Salt stress (200 mM NaCl) caused a significant over-expression of the six selected genes including two ion transporters (high-affinity potassium channel and Na^+^/K^+^ ion exchanger, HKT1; NKX), one Na^+^ exclusion enhancer (ENH1), two transcription factors (bZIP-60, MYB-35), and one vacuolar proton ATPase sub-unit H-like (VPAH) in the Oscar plants. Over-expression of six genes validated the transcriptome data for DEGs (Figure 6). In addition, the correlation between RNA-seq fold change of genes and fold change of qRT-PCR data is r^2^ = 0.731.

## 4. Discussion

Developing salt-tolerant crop cultivars is limited mainly due to a poor understanding of the detailed mechanism of salt tolerance and exact cellular sites of salt damage. Thus, an assessment of the physiological and molecular basis of salt tolerance was carried out in canola—a potential oilseed crop [3,14,33]. In this study, selected canola cultivars were evaluated for salt tolerance at the adult vegetative growth stage. Though all six canola cultivars maintained their degree of salt tolerance, canola cultivars DGL and Dunkled were the most salt tolerant, while cv. Oscar was found to be the most salt-sensitive.

### 4.1. Physiological Basis of salt Tolerance -Water Status, Ion Exclusion, and Antioxidant Potential

For selecting plants for enhanced salt tolerance, selection should be carried out using the most suitable and consistent selection criteria [2,34]. The suitability of any biochemical or physiological indicator for salt tolerance in any species can be suggested if it is highly associated with biomass production [35]. Maintenance of plant water status is essential for the regulation of cell turgor, cell elongation, and cell division—plant growth. Water status in canola cultivars was measured as relative water content (RWC) or water potential (WP). Canola cultivars differed in maintaining RWC under salt stress. However, the degree of salt tolerance in canola cultivars cannot be related to leaf RWC. These results are in contrast to what has been earlier found that leaf RWC can be used as a potential selection criterion such as in sunflower [36], *Vigna radiata* [37], sorghum [38], and rice [39].

In the present study, salt-tolerant cultivars had lower Na^+^, but higher K^+^ accumulation in their leaves than that in the salt-sensitive cultivars. The highest Na^+^ accumulation and lowest K^+^ in the leaves of salt-stressed plants were found in the most salt-sensitive canola cultivar Oscar. These results can be explained in view of the arguments of various researchers that plants of salt-sensitive cultivars have poor control on the transport of toxic ions such as Na^+^ and Cl^-^ from root to shoot, and greater transport constraint for K^+^ from root to shoot, thus, enabling greater salt sensitivity [40,41,42]. Furthermore, salt-tolerant DGL and Dunkled had the highest K^+^ uptake with minimal Na^+^ transport to the leaves under saline conditions. Moreover, it is suggested that ion exclusion might have a key role in the degree of salt tolerance in the canola germplasm examined in this study.

Salt-induced production of MDA and H_2_O_2_ was higher in the salt-sensitive cultivars (Oscar and Legend) than that in the salt-tolerant cultivars (Dunkled and DGL). Plants up-regulate the activity of various enzymatic and non-enzymatic antioxidants to counteract ROS-induced oxidative stress [43]. For example, salt-induced enhanced activity of enzymatic antioxidants and levels of non-enzymatic antioxidants have been reported earlier in a number of plants, e.g., millet seedlings [44].

### 4.2. Activity and Structural Stability of PSII

Our results for the maximum potential of PSII activity (*F*_v_/*F*_m_) do not support the findings of earlier studies that salt stress inhibits PSII activity [5,11,45,46]. However, the results from the present study can be related to some earlier studies in which salt stress had no effect on PSII activity [12,26,47]. In addition, after two weeks of salt stress, there was no change in the maximum yield of primary photochemistry (Φ_Po_). These results indicated that these parameters did not give sound information about the mechanism of salt action on PSII. However, Dawbrowski and his co-workers [46] demonstrated that salt stress reduced the primary photochemistry in rye plants after three to four weeks. This indicates that this parameter depends on the intensity and duration of salt stress. However, the performance index (PI_ABS_) increased with an increase in salt stress, but after three weeks of salt stress, it declined substantially. These results are similar to those of Kalaji and his co-workers [48], who found that salt stress reduced both the quantum yield of PSII and PI_ABS_ after 28 days of salt stress in young tree plants (*Tilia cordata*). The performance index reflects three independent parameters including the density of active reaction centers (RCs), the probability of PSII RCs trapping the absorbed photons, and the probability that PSII RCs will use trapped photons in electron transport [49]. This parameter also depends on changes in four interconnected attributes ABS/RC, Tro/RC, Eto/RC and Dio/RC. Salt stress reduced the absorption and trapping flux with almost no change in electron transport before 21 days of salt stress. It is presumed that no changes in electron transport in the salt-sensitive cultivar Oscar led to increased ROS and oxidative damage as has been observed earlier [46,48]. The results indicated a smaller impact on primary photochemistry on canola plants before three-week of salt stress.

### 4.3. Molecular Basis of Salt Tolerance (Transcriptome Analysis)

In the present study, the physiological basis of salt stress sensitivity in the salt-sensitive canola cultivar Oscar was further validated by molecular analysis, i.e., transcriptome analysis, which is commonly used to screen candidate genes involved in stress responses [14,15]. However, due to the non-availability of the complete genome of *B. napus*, annotating genes and identification of candidate genes for salt tolerance is one of the major obstacles. Since the expression of three genes was verified using qRT-PCR, it is expected that the global gene expression profile for salt-sensitive canola cultivar Oscar reflects the real inside story.

One of the major reasons for a decline in growth due to salt stress is a decline in the biosynthesis of hormones which is evidenced by the exogenous application of hormones such as auxin, cytokinin, gibberellins, and salicylic acid alleviated the adverse effects of salt stress on plant growth of different crop species such as wheat, rice, canola, maize, etc. [2,10,50]. From enrichment analysis of DEGs related to response to hormones, jasmonate signaling, and auxin-mediated efflux are more significant. In addition, large numbers of DEGs related to auxin, jasmonate, ethylene, and ABA were down-regulated in the salt-sensitive canola cultivar Oscar (Figure 8). These results indicate that hormonal signaling pathways were involved in the salt tolerance of the salt-sensitive cultivar Oscar. These results are analogous to those of some earlier studies in which it was found that signal transduction pathways of ABA, auxin, and jasmonates were highly responsive as observed in alfalfa [51].

As described earlier, plants accumulate organic and inorganic solutes to combat salt-induced osmotic stress. Among organic osmolytes, proline and amino acids are potentially compatible solutes that significantly contribute to the degree of salt tolerance. Salt stress up-regulated DEGs associated proline and arginine metabolism pathways (Figure 6; Appendix A). In addition, salt stress down-regulated three DEGs for the proline degrading enzyme, proline dehydrogenase (Appendix A). Similarly, DEGs for trehalose degrading enzymes trehalase were down-regulated under salt stress in the salt-sensitive cultivar Oscar (Appendix A). Such a pattern of gene regulation for proline and trehalose might have favored the increased accumulation of proline and trehalose. These results are similar to those of Wang et al. [52] who reported that salt stress up-regulated the DEGs related to proline biosynthesis in canola.

Salt stress causes the accumulation of Na^+^ in photosynthetic tissues and impairs chloroplastic functions by inhibiting photochemical reactions and biochemical reactions [11]. Several studies have demonstrated that salt stress caused down-regulation of genes associated with chlorophyll biosynthesis [9,15,16,17], and up-regulated thylakoidal proteins related to light reaction [53]. In this study, several GO terms in biological processes, cellular components, and molecular functions are related to PSII, PSI, and CO_2_ fixation. Among highly expressed DEGs, 88 DEGs out of 135 DEGs related to photosynthesis were down-regulated. Most of the down-regulated DEGs in the salt-sensitive canola cultivar Oscar were related to PSII and PSI. These results can be related to earlier findings [9] in which it was found that salt stress caused the down-regulation of the majority of photosynthetic genes such as *Lhc*s, *Pet*s, *Psa*s, and *Psb*s in a cultivar of alfalfa. Moreover, the reduction in expression of photosynthetic genes was greater in the salt-sensitive alfalfa cultivar Xingjiang Daye (XJ) than those in the salt-tolerant cultivar Zhongmu-1 (ZM) [9]. Similarly, a greater number of DEGs related with photosynthetic machinery were up-regulated in salt-tolerant genotypes of alfalfa [54]. In another study, expression of thylakoidal proteins Psb27, PsaO, PetC, and LHCs increased in the leaves of salt-stressed plants of *Carex rigescence* and they all contributed to salt tolerance [55]. From these reports and the results from this study, it is suggested that down-regulation of a greater number of DEGs related to PSII and PSI indicated photo-inhibition of PSII and PSI occurring in the salt-sensitive canola cultivar Oscar.

In this study, KEGG pathways expression analysis of DEGs showed that among enriched down-regulated DEGs, peroxidase and glutathione metabolism are important groups. In this study, a large number of DEGs related to antioxidant activity were down-regulated (85) in the salt-sensitive canola cultivar Oscar. In this study, 34 DEGs related to glutathione-*S*-transferase, 27 DEGs related to peroxidase, and 16 DEGs related to catalases were found to be down-regulated. These results are similar to those of earlier studies in which it was found that SOD and POD activity-related genes are mainly responsible for antioxidant responses such as in alfalfa [51,56], and rice [57]. Moreover, a greater number of DEGs were down-regulated due to salt stress in the salt-sensitive canola cultivar Oscar. As in the present study, several studies indicated that an increase in the activities of antioxidant enzymes as well as the expression of genes involved in antioxidants is advantageous in adapting to salt stress [51,56,58]. These results suggest that the up-regulation of POD and SOD along with the down-regulation of glutathione-*S*-transferase and catalase play an important role in the salt tolerance of the canola cultivar Oscar.

In this study, salt stress up-regulated GO terms associated with DEGs that are related to ion transport, membrane transport, and vesicle trafficking. Appendix A showed that 68 up-regulated DEGs and 55 down-regulated DEGs were found to be involved with ion transport and maintenance of ion homeostasis. Most of the DEGs are related to transporters and channels of sodium, potassium, and calcium transport. For example, up-regulated 11 DEGs are related to Na^+^/H^+^ antiporters which play a key role in Na^+^ exclusion from cytosol and salt tolerance. Of various Na^+^/H^+^ exchangers, some are plasma-membrane localized and some are vacuolar membrane localized, and over-expression of both genes increased the salt tolerance in glycophytic plants [4]. Expression of high-affinity potassium transporter HKT1 in the root xylem parenchyma or the leaf xylem parenchyma cells helps in controlling the loading and accumulation of sodium in the leaf to reduce toxicity. In this study, 14 DEGs for potassium transport were down-regulated. The extent of sodium exclusion and potassium uptake and translocation from root to shoot is linked with Ca^2+^-mediated signaling pathways [59,60]. For example, calcinurin B like 4 (CBL4; SOS3) senses the cytosolic rise in Ca^2+^ due to salt stress by interacting with calcinurin interacting protein kinase (CIPK24), thereby activating the plasma-membrane localized Na^+^/H^+^ antiporter by phosphorylation [4,59,61]. Calcium binding proteins, calcium calmodulin (CaM) proteins, negatively regulated the salinity stress tolerance in barley by down-regulating HKT1;5 and/or up-regulated HKT1;1 [62,63]. This is in line with the results from the present study that 25 down-regulated DEGs are related to calmoudlin binding proteins and calcium-binding proteins in the salt-sensitive canola cultivar Oscar. While working with salt-tolerant and salt-sensitive cultivars of barley, Zhu and their workers [62] reported that 43 DEGs related calmodulin and calmodulin-like proteins were down-regulated in the salt-sensitive barley cultivar N33. These reports and the results from the present study suggest that salt tolerance in canola was associated with Na^+^ loading in the root xylem via Ca^2+^ signaling (Figure 9).

Transcription factors (TFs) play a key role in regulating physiological and developmental processes. In the present study, NAC, MYB, and bZIP transcription factors were enriched in the salt-sensitive canola cultivar Oscar indicating their major role in reprogramming plant metabolism. These results are somewhat similar to earlier studies. For example, while working with the seedlings of *Brassica napus* L. [64] it was found that both MYB and NAC TFs were up-regulated due to salt and drought stress. Similarly, Johnson, et al. reported that the ABA-response element binding factor (ABF), a member of the bZIP family of TFs, was up-regulated in a salt-sensitive wheat cultivar. These results and already published reports mentioned above indicated that these TFs might have played a key role in the salt tolerance of the salt-sensitive canola cultivar Oscar.

## 5. Conclusions

A considerable genetic variability for salt tolerance has been found in local and exotic cultivars of canola, which is associated with the accumulation of potassium, K^+^/Na^+^ ratio, and higher photosynthetic efficiency and antioxidant activity. Cultivars Dunkled, DGL, and Faisal Canola were found salt tolerant, whereas Cyclone, Oscar, and Legend were found to be salt-sensitive. Transcriptome analysis of salt-sensitive canola cultivars revealed that down-regulation of photosynthesis, secondary metabolism, hormones, and antioxidants were found to be associated with a degree of salt sensitivity in cv. Oscar. From this information and further analysis for gene-networking and protein–protein interactions, candidate genes for salt stress sensitivity can be identified. Such information will be helpful in devising strategies to develop salt-tolerant cultivars.

## Figures and Tables

**Figure 1 genes-14-00003-f001:**
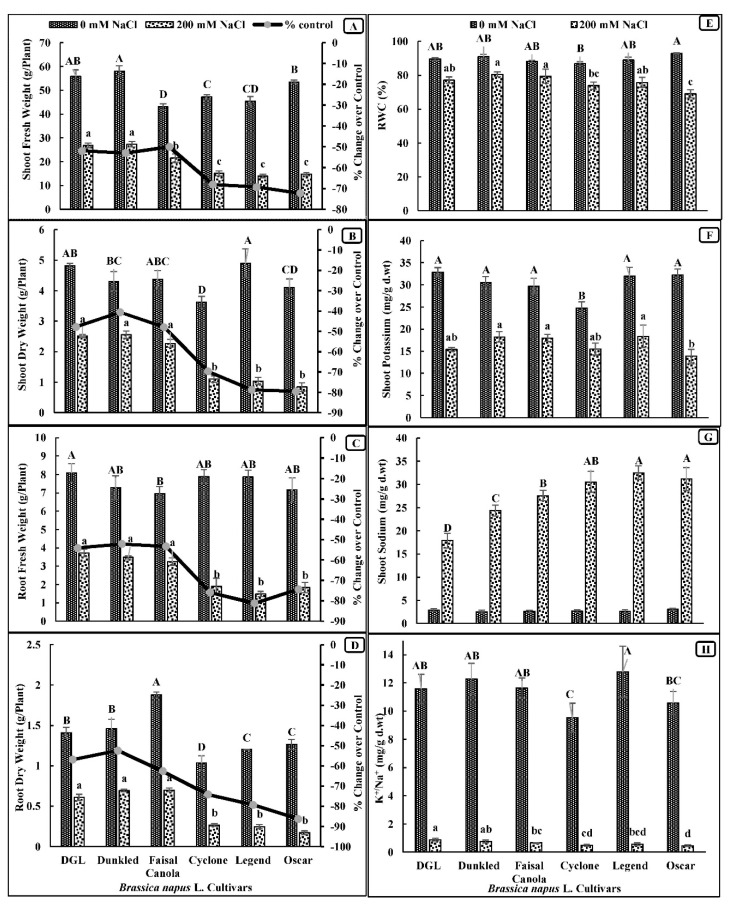
Growth and ion accumulation in seven-week-old plants of canola when three-week-old plants were subjected to salt stress (0 mM and 200 mM NaCl) for four weeks. (**A**) Shoot fresh weight. (**B**) Shoot dry weight. (**C**) Root fresh weight. (**D**) Root dry weight. (**E**) Relative water content. (**F**) Shoot Na^+^. (**G**) Shoot K^+^. (**H**) Shoot K^+^/Na^+^ ratio. *n* = 4. Means were compared with LSD_0.05_ of the interaction term (Salinity × Cvs) only. Means with the letters (A,B) for control and (a,b) for saline differed significantly at 5% probability.

**Figure 2 genes-14-00003-f002:**
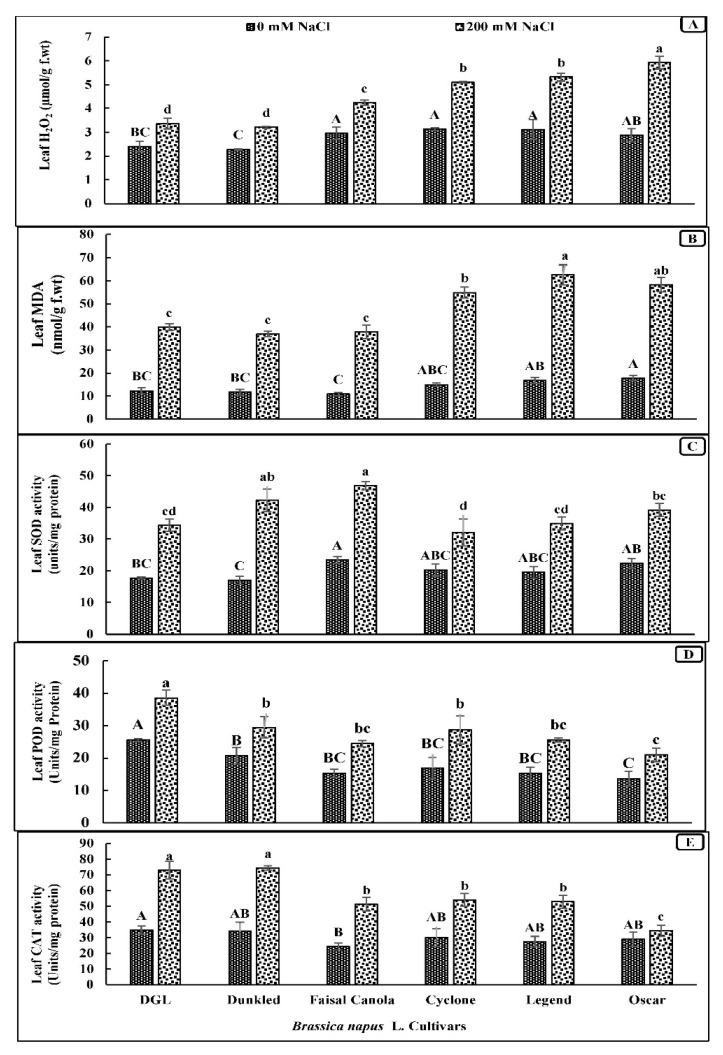
Production of ROS and antioxidative enzyme activity in the leaves of canola cultivars when three-week-old plants were subjected to salt stress (0 mM and 200 mM NaCl). (**A**) Leaf H_2_O_2_. (**B**) Leaf MDA. (**C**) Leaf SOD. (**D**) Leaf POD. (**E**) Leaf CAT activity. *n* = 4. Means were compared with LSD_0.05_ of the interaction term (Salinity × Cvs) only. Means with the letters (A,B) for control and (a,b) for saline differed significantly at 5% probability.

**Figure 3 genes-14-00003-f003:**
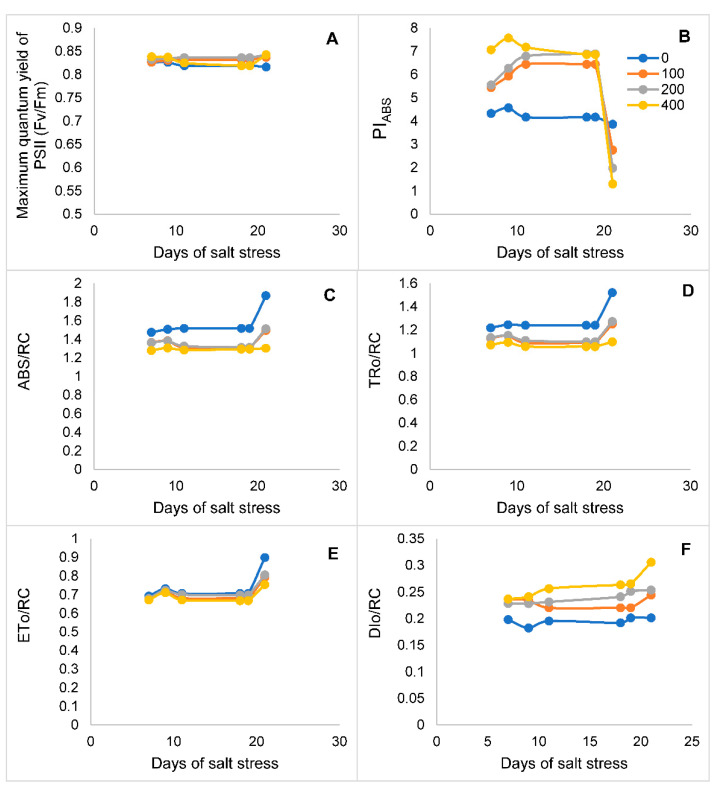
JIP-test parameters of canola cultivars when two-week-old plants of canola cultivar Oscar were subjected to salt stress (0, 100, 200, and 400 mM NaCl). (**A**) Maximum quantum yield of PSII. (**B**) Performance index (PI_ABS_). (**C**) Absorption flux (ABS/RC). (**D**) Trapping flux (TRo/RC). (**E**) Electron transport flux (ETo/RC). (**F**) Heat dissipation flux (DIo/RC).

**Figure 4 genes-14-00003-f004:**
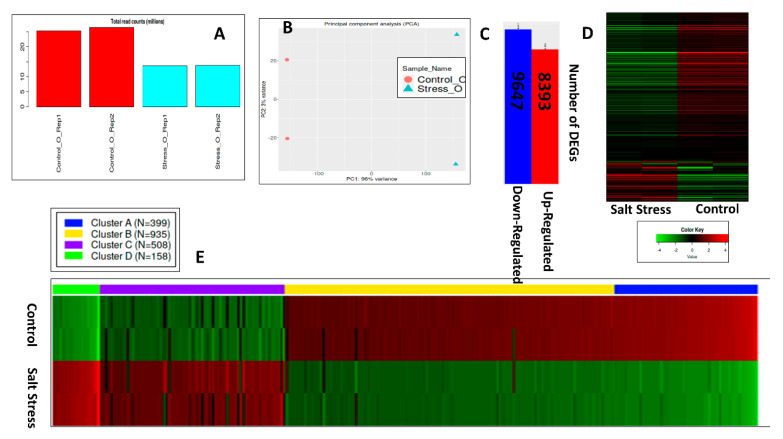
Identification of DEGs in cultivar Oscar under control and salt stress. (**A**) Read counts per million in four samples; (**B**) principal component analysis of RCPM; (**C**) Number of up- and down-regulated genes; (**D**) heat map displaying the clustering analysis of DEGs in up- and down-regulation; (**E**) heat map displaying the expression of DEGs based on k-means enrichment analysis.

**Figure 5 genes-14-00003-f005:**
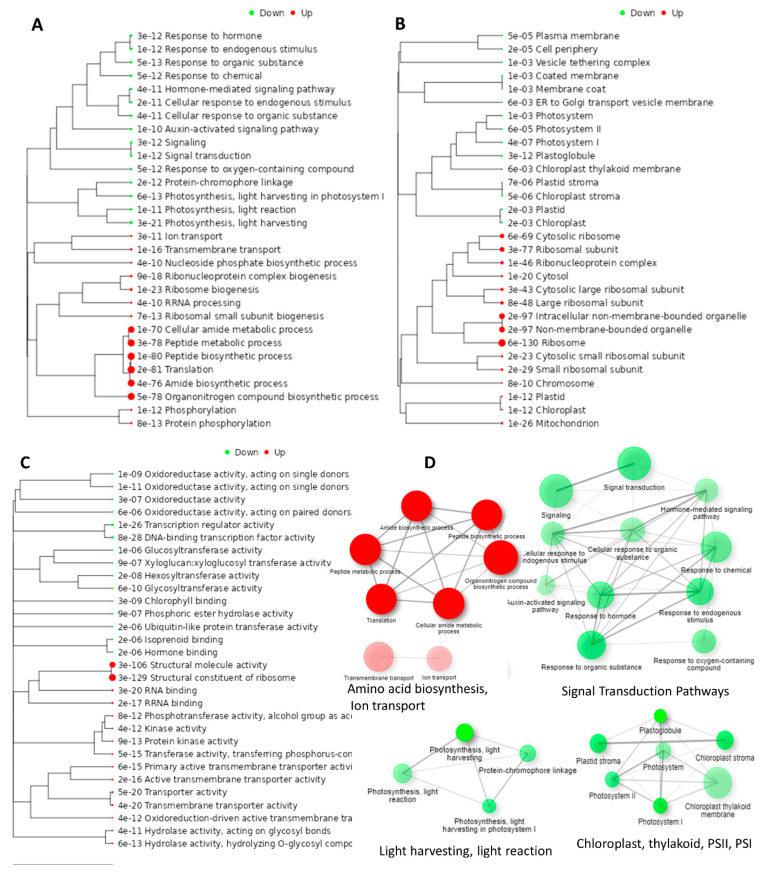
Hierarchical clustering tree of up-regulated and down-regulated DEGs enriched in gene ontology (GO) terms in canola (*B. napus* L.) cultivar Oscar when subjected to salt stress. The 2000 most variable genes were included with number of clusters normalized by gene-mean center. (**A**) GO biological processes; (**B**) GO cellular processes; (**C**) GO molecular functions associated with DEGs; and (**D**) network interaction analysis of GO terms that were up-regulated DEGs (Red nodes and edges) and down-regulated DEGs (green nodes and edges) enriched in GO biological process, cellular component and molecular functions.

**Figure 6 genes-14-00003-f006:**
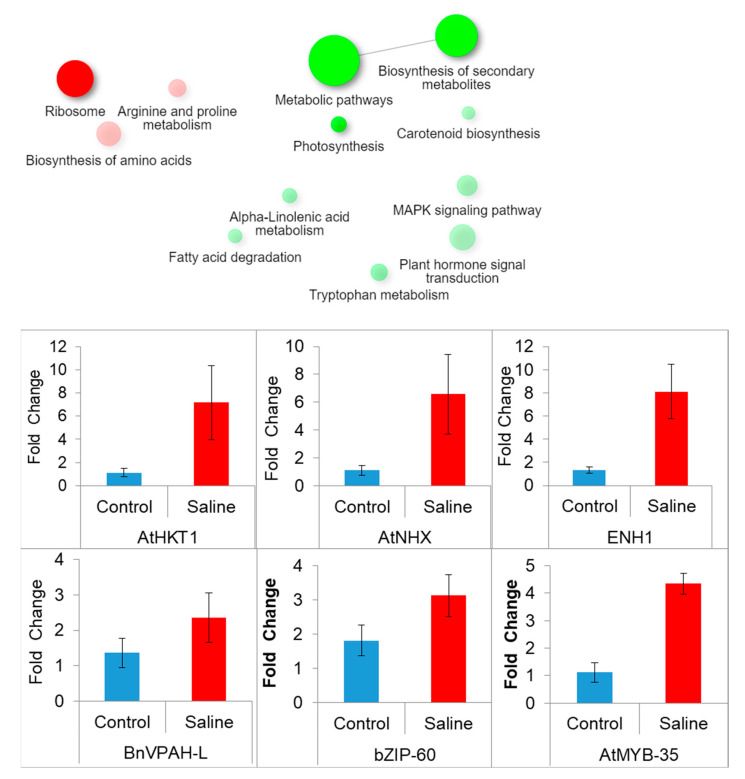
Enriched KEGG pathways up-regulated (*red*) and down-regulated (*green*) in canola (*B. napus* L.) grown under salt stress. Gene expression as fold change of six genes from leaf tissue of canola (*B. napus* L.) salt-sensitive cultivar Oscar subjected to salinity stress (200 mM NaCl). Primer sequences are given in Appendix A.

**Figure 7 genes-14-00003-f007:**
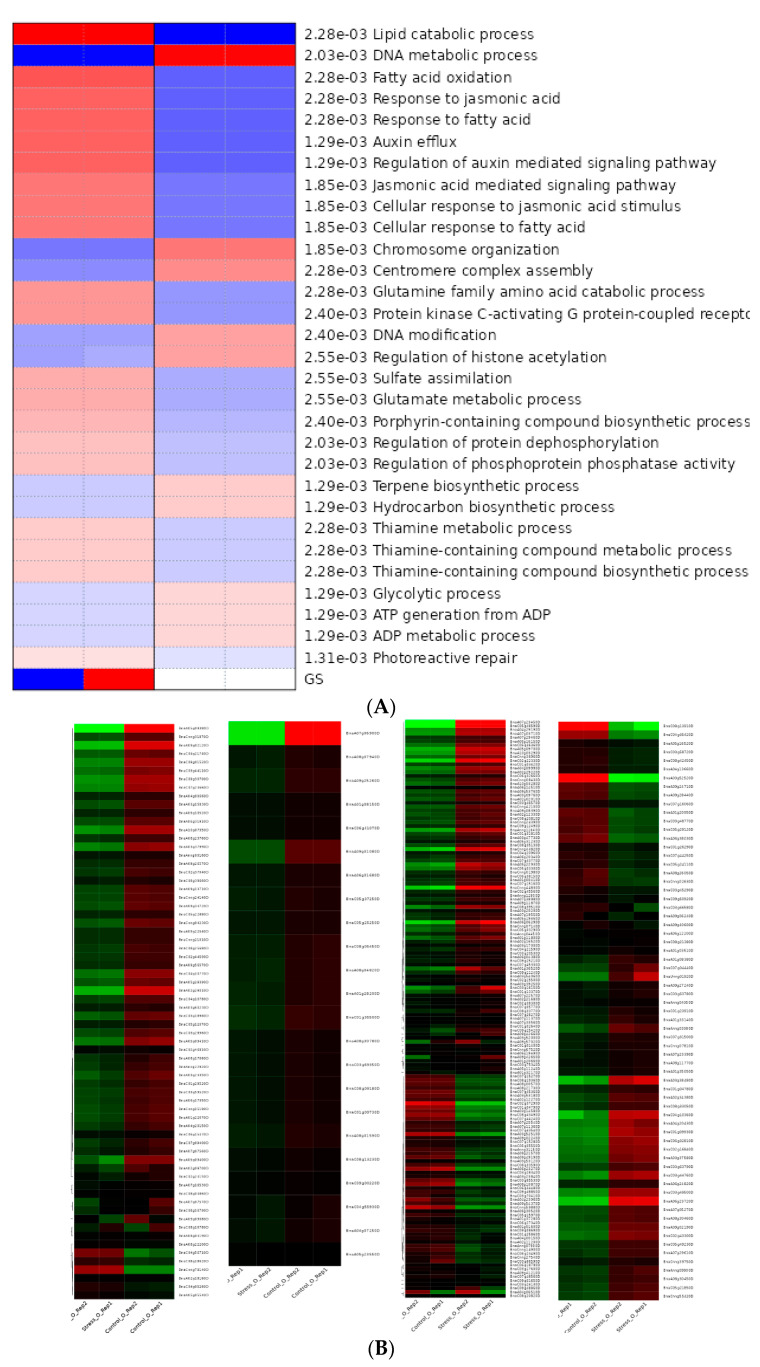
KEGG expression profiles of DEGs using PGSEA. (**A**) KEGG terms associated with biological functions of DEGs using PGSEA; (**B**) list of genes up-regulated or down-regulated associated with PSII, PSI, and peroxidase jasmonate signaling. Red and green indicate up-regulated and down-regulated genes, respectively.

**Figure 8 genes-14-00003-f008:**
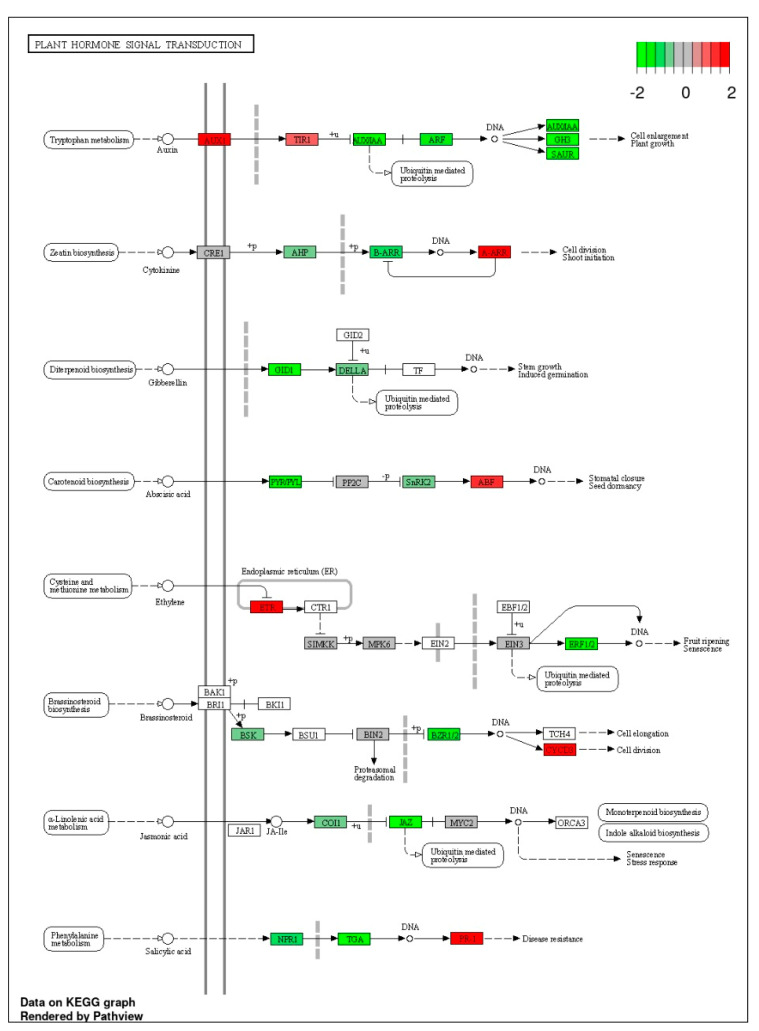
Expression profile of hormonal signaling-pathway-related genes visualized on KEGG Pathway using “Pathview”. The proteins marked green were down-regulated, while that of red were up-regulated under salt stress conditions in the salt-sensitive canola cultivar Oscar.

**Figure 9 genes-14-00003-f009:**
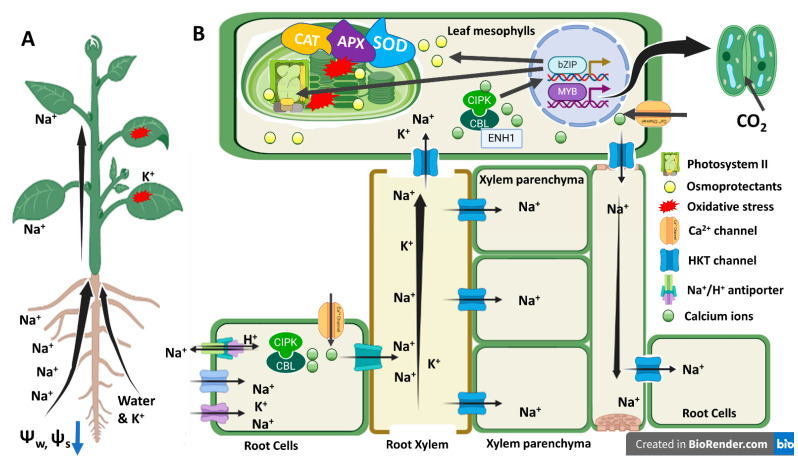
Physiological and molecular mechanism of salt tolerance in canola. (**A**) Salt stress increases Na^+^ uptake and translocation from root to shoot, while it reduces water and K^+^ uptake. This causes nutritional imbalance and oxidative stress. (**B**) Na^+^ uptake at the root level occurs via potassium channels. Xylem parenchyma unloads Na^+^ from xylem via HKT channel. Plant cells sense Na^+^ via Ca^2+^ channels which activate downstream hormonal signaling pathways, photosynthesis, and antioxidant potential.

**Table 1 genes-14-00003-t001:** Mean squares from the analysis of variance (ANOVA) of the data for growth attributes, relative water content, accumulation of K^+^ and Na^+^, H_2_O_2_ as reactive oxygen species (ROS), membrane damage as oxidative stress, and activities of antioxidant enzymes of six cultivars of canola (*B. napus* L.) grown under control (0 mM NaCl) or saline (200 mM NaCl) conditions.

Source of Variation	Mean Squares			
	Cultivar	Salinity	Cultivar × Salinity	Error
Degree of Freedom (df)	5	1	5	36
Attributes				
Shoot Fresh Weight	240.71 ***	11,249.359 ***	61.29 ***	6.643
Root Fresh Weight	2.105 **	292.152 ***	2.604 ***	0.485
Shoot Dry Weight	2.256 ***	83.136 ***	1.234 ***	0.15
Root Dry Weight	0.510 ***	10.303 ***	0.063 ***	0.009
Relative Water Content	36.555 *	2585.238 ***	64.675 **	14.069
Shoot Potassium Content	29.10 **	2324.88 ***	21.815 *	6.573
Shoot Sodium Content	59.68 ***	7272.54 ***	60.669 ***	4.764
Na^+^/K^+^ Ratio	3.228 ns	1390.26 ***	2.453 ns	1.965
Leaf Hydrogen Peroxide	4.04 ***	36.02 ***	1.39 ***	0.13
Leaf Malondialdehyde	403.63 ***	14,145.74 ***	154.22 ***	13.58
Leaf Superoxide Dismutase	98.11 ***	3989.43 ***	52.02 **	13.92
Leaf Peroxidase	215.92 ***	1218.06 ***	8.45 ns	17.35
Leaf Catalase	655.72 ***	8545.44 ***	310.73 ***	52.81

*, **, *** were significant at 0.05, 0.01 and 0.001 probability; ns = non-significant.

## Data Availability

The raw transcriptome data were submitted at NCBI and available at https://www.ncbi.nlm.nih.gov/bioproject/558913 accessed on 1 October 2022.

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
