# Peer review of "Photosynthesis and Salt Exclusion Are Key Physiological Processes Contributing to Salt Tolerance of Canola (Brassica napus L.): Evidence from Physiology and Transcriptome Analysis"

_genes, 2022, doi:10.3390/genes14010003_

Round 1
Reviewer 1 Report
1. The authors have mentioned two treatments (0 and 200mM) in the experimental section. However, i see in JIP test authors have mentioned 4 treatments 0, 100, 200 and 400mM. Line 336. Kindly mention the right treatment methods in the experimental section.
2. I see the authors use only 4 replicates. How did the authors achieve statistical significance? Authors should mention what kind of ANOVA used in the experimental section.
3. Figure 5 is not clear. Kindly replace the Figure 5 with a higher resolution.
4. Some references don't have DOI:, 10, 23, 36, 37, 40, 43, 74, 75, 76
Author Response
Hazel He
Section Managing Editor
Email: [email protected]
Genes | Special Issue Mentor Program
Subject: Submission of revised version of the MS, Photosynthesis and Salt Exclusion are Key
Physiological Processes Contributing in Salt Tolerance of Canola (Brassica napus L.): Evidence from Physiology and Transcriptome Analysis.
Reference: Manuscript ID: genes-2024489
Dear Sir /Madam
With reference to comment raised by the reviewers on our MS. Thank you very much for giving sufficient time to revise the MS. The MS has been revised keeping in view of the comment raised by the reviewers. Our responses to the comments are as follows:
Reviewer 1
In this study, authors carried out the six canola cultivars for salt tolerance by subjecting 0 or 200 mM NaCl stress. I found this topic interesting but I have many concerns related to the research article. In my opinion, Introduction and discussion are well-written. However, aim and objective of the study are not clearly mentioned. Part of the methods section requires detailed information. Some of the sentence structure and phrasing are weak throughout the manuscript. The result section requires a thorough review. Discussing the results further would help to strengthen the manuscript.
Response: This reviewer critically read the MS and raised several critical but constructive comments. These comments are very helpful in improving the quality of MS. So, we considered of all comments while revising the MS.
- Line 92-94. “Canola seed is ... linolenic acid.” This sentence does not serve any purpose and should be deleted.
Response: This sentence has been deleted as suggested by the reviewer.
- Line 105-119, This paragraph should be subtitled, "Plant material".
Response: The first paragraph was placed under sub-title Plant Material and the number of all other sub-sequent headings were changed accordingly.
- Line 232, accession number PRJNA687396 and PRJNA558913, Are these data composed of two accession numbers? Please make it clear.
Response: Thanks for pointing-out a very critical error. Project number has been rectified as PRJNA558913.
- In the legend of Figure 1, the meaning of the letter C/c and D/d are not clearly explained.
Response: Thank you very much for highlighting this deficiency. Cultivar means were compared under control and saline conditions. Now this has been given in the Figure legend as means with the letters (A-B) for control and (a-b) for saline differed significantly at 5% probability.
- There are many inappropriate title formats in this article. For example, 3.7.1 and 3.7.2 can be combined. Please check the entire manuscript and make corresponding changes.
Response: Thank you very much for critical but positive suggestions. The sub-titles 3.7.1 and 3.7.2 were combined and subsequent headings were numbered accordingly. In addition, several sub-title headings were deleted as pointed out by the reviewer.
- Figure 6, only 3 DEGs used for qRT-PCR, it's not enough for validation.
Response: Thank you very much for highlighting this deficiency. Now data of qRT-PCR of six genes were presented. Figure 6 has been revised now in view of the comments raised by the reviewers. The primer used for additional genes were also given in the Supplementary Table 3.
- In the 2.9.1, Did the authors perform transcriptome sequencing on “Oscar”'s leaves of two salt treatments (0 and 200 mM NaCl)? This is not clear, please refine it. In addition, why did the authors choose “Oscar” species for transcriptome sequencing, and is it different from other species in terms of physiological growth and other indicators?
Response: Thank you very much for pointing out this error. The canola cultivar Oscar is high yielding cultivar with greater number of pods and seed size. However, this is ranked as salt sensitive. Assessment of molecular basis of salt stress sensitivity and introgression of those traits in to this high yield cultivar is the major objective. So, cv. Oscar was used for transcriptome analysis. Now briefly explained how salt stress was applied to the cv. Oscar as suggested by the reviewer.
- The title of TABLE 1 is wrong. It should belong to the results section.
Response: Thank you very much. The title of the Table 1 has been amended and order of appearance of physiological and biochemical attributes were corrected as suggested by the reviewer. The appearance of the Table 1 is now after heading RESULTS as suggested by the reviewer.
- The authors have classified the functions of DEGs, but only in a superficial way. In my opinion, these DEGs should be presented in the form of heatmaps or pathways, which can better show the genes related to salt resistance.
Response: Thank you very much for your kind suggestion. Bioinformatic analysis was carried out at iDEP and data for exploratory analysis, GO and KEGG pathway analysis, expression profile of DEGs as heat map, KEGG pathway visualized at Pathview are included now.
- The authors did not explain the analysis method of KEGG pathway in the manuscript.
Response: Thank you very much for pointing out this deficiency in the manuscript. The description of KEGG pathway has been described in M&M section as suggested by the reviewer.
- Graphic quality should be improved, such as Figure 5.
Response: Thank you very much. The graphic quality of Fig 5 is low as this has been downloaded from web-based bioinformatics tool. However, the quality of figure and legends have been improved as much as we can as suggested by the reviewer.
- It would be helpful to deepen the readers' understanding if a schematic diagram of canola regulating salt stress could be added.
Response: Thank you very much for highlighting this deficiency. Now a schematic diagram has been drawn of good quality. This schematic diagram can be used as Graphical Abstract as well as suggested by the reviewer.
Hope you will find revised version of the MS suitable for publication in the esteemed Journal. However, if you find any error, please let us know for its rectification.
Best Regards
Habib-ur-Rehman Athar
Reviewer 2 Report
In this study, authors carried out the six canola cultivars for salt tolerance by subjecting 0 or 200 mM NaCl stress. I found this topic interesting but I have many concerns related to the research article. In my opinion, Introduction and discussion are well-written. However, aim and objective of the study are not clearly mentioned. Part of the methods section requires detailed information. Some of the sentence structure and phrasing are weak throughout the manuscript. The result section requires a thorough review. Discussing the results further would help to strengthen the manuscript.
1. Line 92-94. “Canola seed is ... linolenic acid.” This sentence does not serve any purpose and should be deleted.
2. Line 105-119, This paragraph should be subtitled, "Plant material".
3. Line 232, accession number PRJNA687396 and PRJNA558913, Are these data composed of two accession numbers? Please make it clear.
4. In the legend of Figure 1, the meaning of the letter C/c and D/d are not clearly explained.
5. There are many inappropriate title formats in this article. For example, 3.7.1 and 3.7.2 can be combined. Please check the entire manuscript and make corresponding changes.
6. Figure 6, only 3 DEGs used for qRT-PCR, it's not enough for validation.
7. In the 2.9.1, Did the authors perform transcriptome sequencing on “Oscar”'s leaves of two salt treatments (0 and 200 mM NaCl)? This is not clear, please refine it. In addition, why did the authors choose “Oscar” species for transcriptome sequencing, and is it different from other species
in terms of physiological growth and other indicators?
8. The title of TABLE 1 is wrong. It should belong to the results section.
9. The authors have classified the functions of DEGs, but only in a superficial way. In my opinion, these DEGs should be presented in the form of heatmaps or pathways, which can better show the genes related to salt resistance.
10. The authors did not explain the analysis method of KEGG pathway in the manuscript.
11. Graphic quality should be improved, such as Figure 5.
12. It would be helpful to deepen the readers' understanding if a schematic diagram of canola regulating salt stress could be added.
Author Response
Hazel He
Section Managing Editor
Email: [email protected]
Genes | Special Issue Mentor Program
Subject: Submission of revised version of the MS, Photosynthesis and Salt Exclusion are Key
Physiological Processes Contributing in Salt Tolerance of Canola (Brassica napus L.): Evidence from Physiology and Transcriptome Analysis.
Reference: Manuscript ID: genes-2024489
Dear Sir /Madam
With reference to comment raised by the reviewers on our MS. Thank you very much for giving sufficient time to revise the MS. The MS has been revised keeping in view of the comment raised by the reviewers. Our responses to the comments are as follows:
Reviewer 2
1-Improve the English language and shorten the whole manuscript from abstract to the end.
Response: Thank you very much for your critical but constructive comments. The whole manuscript has been critically read and shorten the MS including the Abstract as suggested by the reviewer.
2- The authors used the fold change method to investigate gene expression. Why is the expression amount for the control sample different?
Response: Thank you very much for clarification of method used. In this study, data for relative changes in gene expression is presented as fold change in expression of target genes in the leaves of salt stressed plants normalized to internal control (β-Actin) and relative to leaves of non-salinized or control plants. Results of RT-PCR was presented at CT values i.e. threshold cycle number of PCR at which amplified product was first detected. The average CT values for both target genes and that of internal control was calculated and minus from triplicate values. The least difference for internal control was used for further calculation. Expression amount of target genes for control sample was also presented with reference to internal control.
3-There is a contradiction. The author explained the relative expression of genes in line 257 (material and method) and the fold change method, but those are different from each other. The 2-ΔΔCt formula is for fold change and the 2-ΔCt method is for the relative gene expression (RGE). please check the Kianersi et al, 2020(Scientific report).
Response: Thank you very much for pointing this technical error and misunderstanding. Now this has been rectified in M&M. In this study, changes in gene expression is presented as fold change in expression of target genes in the leaves of salt stressed plants normalized to internal control (β-Actin) and relative to leaves of non-salinized or control plants.
4- Add more details and discussion about the amount of expression of 3genes in the text. What was the influential factor for changes?
Response: Thank you very much for pointing it out. Now data of qRT-PCR of six genes were presented. Figure 6 has been revised now in view of the comments raised by the reviewers. The primer used for additional genes were also given in the Supplementary Table 3. The Discussion section has been revised and discussed their influential role as suggested by the reviewer.
Hope you will find the revised version of the MS suitable for its publication in your valuable Journal Genes.
However, still if you find some errors in it, please don’t hesitate to contact us for its rectification.
Best Regards
Dr. Habib-ur-Rehman Athar
Institute of Botany
Bahauddin Zakariya University, Multan Pakistan
Reviewer 3 Report
1-Improve the English language and shorten the whole manuscript from abstract to the end.
2- The authors used the fold change method to investigate gene expression. Why is the expression amount for the control sample different?
3-There is a contradiction. The author explanted the relative expression of genes in line 257 (material and method) and the fold change method, but, those are different from each other. The 2-ΔΔCt formula is for fold change and the 2-ΔCt method is for the relative gene expression (RGE). please check the Kianersi et al, 2020(Scientific report).
4- Add more details and discussion about the amount of expression of 3genes in the text. What was the influential factor for changes?
.
Author Response
Hazel He
Section Managing Editor
Email: [email protected]
Genes | Special Issue Mentor Program
Subject: Submission of revised version of the MS, Photosynthesis and Salt Exclusion are Key
Physiological Processes Contributing in Salt Tolerance of Canola (Brassica napus L.): Evidence from Physiology and Transcriptome Analysis.
Reference: Manuscript ID: genes-2024489
Dear Sir /Madam
With reference to comment raised by the reviewers on our MS. Thank you very much for giving sufficient time to revise the MS. The MS has been revised keeping in view of the comment raised by the reviewers. Our responses to the comments are as follows:
Reviewer 3
- The authors have mentioned two treatments (0 and 200 mM) in the experimental section. However, I see in JIP test authors have mentioned 4 treatments 0, 100, 200 and 400 mM. Line 336. Kindly mention the right treatment methods in the experimental section.
Response: Thank you for highlighting this deficiency. In M&M section, the method under sub-title 2.7 Assessment of photosystem-II (PSII) structural stability using JIP-test was revised as
Two-week old plants of salt sensitive cultivar Oscar were subjected to varying levels of salt stress (0, 100, 200, 400 mM NaCl) in half strength of Hoagland’s nutrient solution for three weeks. After one week of salt stress, data of OJIP transients was obtained using hand-held chlorophyll fluorescence meter (FluorPen FP100, Photon Systems Instruments, Czech Republic). Line 186-190
- I see the authors use only 4 replicates. How did the authors achieve statistical significance? Authors should mention what kind of ANOVA used in the experimental section.
Response: Thank you very much for highlighting this deficiency. The details of statistical analysis of two-way ANOVA have been given in M&M section.
- Figure 5 is not clear. Kindly replace the Figure 5 with a higher resolution.
Response: Thank you very much. The graphic quality of Fig 5 is low as this has been downloaded from web-based bioinformatics tool. However, the quality of figure and legends have been improved as much as we can as suggested by the reviewer.
- Some references don't have DOI:, 10, 23, 36, 37, 40, 43, 74, 75, 76
Response: Thank you very much for point out this error in references. We use EndNote 20 for reference management and citation using MDPI citation style format. However, during import of reference some time doi numbers did not become imported. Thank you very much. We have inserted the doi numbers by updating our library. However, for reference 36 and 43, the Pak J Bot did not insert doi numbers on old papers yet.
Hope you will find the revised version of the MS suitable for its publication in your valuable Journal Genes.
However, still if you find some errors in it, please don’t hesitate to contact us for its rectification.
Best Regards
Dr. Habib-ur-Rehman Athar
Institute of Botany
Bahauddin Zakariya University, Multan Pakistan
Round 2
Reviewer 2 Report
Based on my suggestion in the first review, the authors have addressed my concerns appropriately. I still have a few questions for the manuscript.
1. Line 357, “Out of top 17000 DEGs”, please explain how to choose 17000 degs?
2. In Figure 5, the legend of Figure 5D is not explained.
3. In Figure 7, there are two B in the legend.
4. In 3.8, the authors said the accuracy of RNA-seq was validated by qRT-pcr, the correlation analyses between RNA-SEQ and qRT-PCR should be performed.
5. Line 616-618, as mentioned by the authors, transcription factors play a key role in regulating plant physiology and development, including MYB, bZIP, bHLH, AP2/ERF, WRKY. I suggested that the authors identify the first few families of transcription factors involved in salt stress, and perform the expression levels of these genes, to better explain the molecular mechanisms.
Author Response
Hazel He
Section Managing Editor
Email: [email protected]
Genes | Special Issue Mentor Program
Subject: Submission of revised version of the MS, Photosynthesis and Salt Exclusion are Key
Physiological Processes Contributing in Salt Tolerance of Canola (Brassica napus L.): Evidence from Physiology and Transcriptome Analysis.
Reference: Manuscript ID: genes-2024489
Dear Sir /Madam
With reference to comment raised by the reviewers on our MS. Thank you very much for giving sufficient time to revise the MS. The MS has been revised keeping in view of the comment raised by the reviewers. Our responses to the comments are as follows:
Reviewer 2
- Improve the English language of the MS.
Response: Thank you very much for your critical but constructive comments. The whole manuscript has been critically read and rectified in track changes by the senior author Prof Dr M Ashraf as suggested by the reviewer.
- Based on my suggestion in the first review, the authors have addressed my concerns appropriately. I still have a few questions for the manuscript. Line 357, “Out of top 17000 DEGs”, please explain how to choose 17000 DEGs?
Response: Thank you very much for highlighting this error. Total DEGs were 18040 with FDR 0.1 and fold change 2 cut off values. It was miss understood that 9647+8393 = 17000. The sum of both up- and down-regulated DEGs is 18040. Now this has been rectified as pointed out by the reviewer.
- In Figure 5, the legend of Figure 5D is not explained.
Response: Thank you very much for pointing it out. Now Figure 5D has explained in the Figure legend as suggested by the reviewer.
- In Figure 7, there are two B in the legend.
Response: This has been rectified by deleting second B in the figure legend as highlighted by the reviewer.
- In 3.8, the authors said the accuracy of RNA-seq was validated by qRT-PCR, the correlation analyses between RNA-SEQ and qRT-PCR should be performed.
Response: The correlation analysis between fold change from RNA-seq data and that from qRT-PCR was performed and found that it was r2 = 0.731. Now this has been mentioned in the MS.
- Line 616-618, as mentioned by the authors, transcription factors play a key role in regulating plant physiology and development, including MYB, bZIP, bHLH, AP2/ERF, WRKY. I suggested that the authors identify the first few families of transcription factors involved in salt stress, and perform the expression levels of these genes, to better explain the molecular mechanisms.
Response: The discussion section related to transcription factors has been revised and only limited to bZIP and MYC. Discussion related to other transcription factors has been deleted now.
Hope you will find the revised version of the MS suitable for its publication in your valuable Journal Genes.
However, still if you find some errors in it, please don’t hesitate to contact us for its rectification.
Best Regards
Dr. Habib-ur-Rehman Athar
Institute of Botany
Bahauddin Zakariya University, Multan Pakistan
Reviewer 3 Report
If you compared control(non-treatment) and treatment leaves for each gene, you must do the relative gene expression (RGE) for all, but if not is, your fold change is ok.
I do not understand your mean about once you calculate the control and onother time consider with reference gene!!???
you must consider all samples for each gene with reference gene like actin.
you can not use ct from 2 things.
please revise that.
is it fold or relative if figure?
if it is fold, you have to put 1 for control.
for relative you must change name but the amount is smaller than that.
Author Response
Hazel He
Section Managing Editor
Email: [email protected]
Genes | Special Issue Mentor Program
Subject: Submission of revised version of the MS, Photosynthesis and Salt Exclusion are Key
Physiological Processes Contributing in Salt Tolerance of Canola (Brassica napus L.): Evidence from Physiology and Transcriptome Analysis.
Reference: Manuscript ID: genes-2024489
Dear Sir /Madam
With reference to comment raised by the reviewers on our MS. Thank you very much for giving sufficient time to revise the MS. The MS has been revised keeping in view of the comment raised by the reviewers. Our responses to the comments are as follows:
Reviewer 3
1-If you compared control(non-treatment) and treatment leaves for each gene, you must do the relative gene expression (RGE) for all, but if not is, your fold change is ok.
I do not understand your mean about once you calculate the control and onother time consider with reference gene!!???
you must consider all samples for each gene with reference gene like actin.
you cannot use ct from 2 things.
please revise that.
is it fold or relative if figure?
if it is fold, you have to put 1 for control.
for relative you must change name but the amount is smaller than that.
Response: We are really sorry for our writing and miss understanding or confused statements about calculations of gene expression. In this study, data for gene expression is presented as fold change in the leaves of salt stressed plants normalized to internal control (β-Actin). Results of RT-PCR was presented at CT values i.e. threshold cycle number of PCR at which amplified product was first detected. The average CT values for both target genes and that of internal control β-actin was calculated. Expression of each replicate of each target genes were normalized to internal control β-actin. However, expression of genes in salt stressed leaves were not presented with reference to their expression in control or non-salinized leaves. Therefore, values of expression of genes in control plants were not 1.
Hope you will find the revised version of the MS suitable for its publication in your valuable Journal Genes.
However, still if you find some errors in it, please don’t hesitate to contact us for its rectification.
Best Regards
Dr. Habib-ur-Rehman Athar
Institute of Botany
Bahauddin Zakariya University, Multan Pakistan